# Dimensional Analysis Model of Head Loss for Sand Media Filters in a Drip Irrigation System Using Reclaimed Water

Yaqi Hu [1] , Wenyong Wu [1],*, Honglu Liu [2], Yan Huang [3], Xiangshuai Bi [4] , Renkuan Liao [4] and Shiyang Yin [2]

1    State Key Laboratory of Simulation and Regulation of Water Cycle in River Basin, China Institute of Water Resources and Hydropower Research, Beijing 100048, China; huyq2011@126.com
2    Beijing Hydraulic Research Institute, Beijing 100048, China; liuhonglu@yeah.net (H.L.); yinshiyang@aliyun.com (S.Y.)
3    Hydrology Bureau of Anhui Province, Hefei 230022, China; twoemperors@163.com
4    Department of Land Engineering, College of Land Science and Technology, China Agricultural University, Beijing 100048, China; xiangshuaibi@yeah.net (X.B.); renkuanliao@163.com (R.L.)
*    Correspondence: wenyongwu@126.com

**Abstract:** A new model was developed to predict head loss in sand media filters. Sand filters with six different media using reclaimed water were used to measure head losses at different flow rates in the laboratory. The parameters influencing head losses were considered to be the uniformity coefficient, the effective diameter, the sand mass, the filtration velocity, the pollution load, and the water viscosity. A dimensional analysis method was used to develop the model. A comparison between the predicted and the measured head losses showed close agreement with a correlation coefficient of 91.7%, reaching a significance level of $p < 0.001$. The results showed that the model might give satisfactory predictions within the following range of operational and filter structure parameters: uniformity coefficient 1.48–3.31; effective diameter 0.41–2.1 mm; pollution load 0.0169–4.2049 kg, filtration velocity 0.0038–0.0398 m/s.

**Keywords:** water-saving irrigation; filter; dimensional analysis model; performance test; parameter regulation

## 1. Introduction

The worldwide shortage of water resources has encouraged the use of recycled municipal wastewater [1–3]. In the United States, Israel, China, and Australia, reclaimed water irrigation is used to mitigate agricultural water shortages [4–6]. Surface and subsurface drip irrigation can reduce the exposure of reclaimed water to human populations, reducing the associated health risks; thus, drip irrigation has become an increasingly recommended use for reclaimed water [7–10]. Studies have shown that reclaimed water contains suspended particles with diameters of less than 100 µm [11], which increases the likelihood of clogging the dripper [12–14]. Compared to disc and screen filters, sand media filters have a higher removal efficiency for Total Suspended Solids (TSS), which can reach 33% to 85% [15,16]. Sand media filters thus represent the most important tool to prevent dripper clogging [17]. The migration of suspended particles in sand-filtration media involves sedimentation, inertia, interception, diffusion, and dynamic effects. In reclaimed water, 86% of suspended particles have diameters greater than 14 µm [18]. The suspended particles are mainly removed by sedimentation and interception [19]. The removal efficiency of sand media filters is related to the characteristics of the sand media, such as structure, shape, and size [20]. The aggregation of suspended particles in the surface or filtration layer significantly increases the head loss and shortens the backwash cycle [21]. A number of studies on the hydraulic performance of irrigation sand filters have been conducted with the goal of improving its TSS removal efficiency and the operational efficiency of drip irrigation systems. The dimensional analysis model is widely used in studies [22];

for example, Duran-Ros et al. [23] (Equation (1)) and Elbana et al. [24] (Equation (2)) have proposed models for calculating head loss, and others have proposed theoretical models to characterize the physics of the process [25] (Equation (3)).

$$\frac{vC^{0.50}}{\Delta H^{0.50}} = 270.58 \left(\frac{\rho}{C}\right)^{-0.9870} \left(\frac{\mu}{\Delta H^{0.50} \cdot C^{0.50} \cdot D_p}\right)^{1.0053} \tag{1}$$

In this equation, $v$ is the filtration velocity (m s$^{-1}$), $\mu$ is water viscosity (Pa s), $\Delta H$ is the head loss (Pa), $C$ is the TSS concentration (kg m$^{-3}$), $D_p$ is the filter inlet or the outlet diameter, and $\rho$ is the water density (kg m$^{-3}$).

$$\frac{\Delta H}{d_e C g} = 16.261 \left(\frac{\rho}{C}\right)^{0.89} \left(\frac{V}{d_e^3}\right)^{-0.031} \left(\frac{d_f}{d_e}\right)^{1.087} \tag{2}$$

In this equation, $d_e$ is the sand's effective diameter in m, which refers to the sieve diameter that allows 10% of the total filter sand to pass through during the sieving process, and it reflects the particle size of the filter sand. $d_f$ is the internal diameter of the sand media filter, and $V$ is the volume of water filtered (m$^3$).

$$\Delta H = \frac{34.74 \mu^{0.48} v^{1.52} (1 - \varepsilon_0)^{1.48}}{\rho^{0.48} g \psi^{1.48} d^{1.48} \varepsilon_0^3} L \tag{3}$$

In this equation, $\varepsilon_0$ is the porosity of the sand filter layer, $L$ is the filtration layer thickness (m), $\psi$ is the sphericity coefficient of filtration media, and $d$ is the equivalent diameter of the filter media (mm). The sand's effective diameter ($d_e$) and uniformity coefficient ($UC_s$) are important parameters affecting the performance of the sand media filter (Elbana et al., 2013). $UC_s$ is the ratio between the $d_e$ and $d_{60}$ which refers to the sieve diameter that allows 60% of the total filter sand to pass through the sieve during the sieving process. The higher the $UC_s$, the more rational the filtration media gradation and the better the filtration efficiency [13]. However, no parameters of sand-filtration media were considered in Equation (1). Although the effects of particle diameter and porosity of sand-filtration media were considered in Equations (2) and (3), the effect of $UC_s$ was not considered, and the effect of the concentration of suspended solids was not considered in Equation (3). Other models have used the thickness of the pollutant aggregation layer [26,27] or the mean diameter of suspended solids [18] as parameters. However, because these parameters are difficult to measure, their application in engineering practice is restricted.

The present paper is an attempt to develop a hydraulic model of a sand media filter using dimensional analysis through testing the hydraulic performance of the sand media filter and considering key parameters such as $d_e$, $UC_s$, and the effects of the suspended particle pollution load of the reclaimed water, and comparing it with the models developed by Duran-Ros et al. [23], Elbana et al. [24], and Dong [25] to verify the model's adaptability.

## 2. Materials and Methods

The tests were conducted at the Huangcun Sewage Treatment Plant in the Daxing district of Beijing; the Orbal secondary biochemical oxidation ditch process was adopted. The TSS concentration after secondary treatment was 3.5 to 88 mg L$^{-1}$, with a mean value of 21.4 mg L$^{-1}$. The sand media filters (model SS-400 × 50) used were manufactured by Beijing Tongjie Company (Beijing, China) with a tank body diameter of 40 cm, a water inlet diameter of 50 mm, and a designed range of flow rate of 5.0–18 m$^3$/h. As shown in Figure 1, three sand media filters were installed in the testing system. A pressure sensor (accuracy: 0.001 MPa) was installed before and after each sand media filter. A flow sensor (accuracy: 0.1 m$^3$) was installed at each filter inlet. The system flow rate and pressure data were collected in real time every 5 min using a computer.

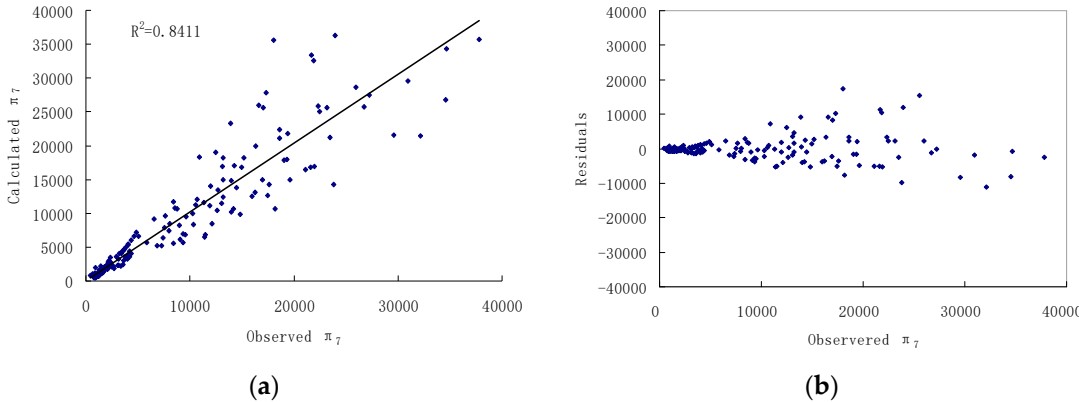

(**a**)　　　　　　　　　　　　　　　　　　　　(**b**)

**Figure 1.** The measured value of $\pi_7$, the calculated value from Equation (6) (**a**), and residues (**b**). $p < 0.05$.

As shown in Table 1, six quartz-sand-filtration media gradations were set up in the test. The test was divided into two sessions. The thickness of quartz sand in the tank (L) was 40 cm.

**Table 1.** Quartz sand parameters in the gradation treatment.

| Treatment | Sand Mass $m$, kg | Porosity $\varepsilon$ | Equivalent Diameter $d$, mm | $d_e$, mm | $d_{60}$, mm | $UC_s$ |
|---|---|---|---|---|---|---|
| Treatment 1 | 79.1 | 0.406 | 2.94 | 2.1 | 3.1 | 1.48 |
| Treatment 2 | 76.2 | 0.428 | 2.02 | 1.41 | 2.20 | 1.56 |
| Treatment 3 | 75.5 | 0.433 | 0.65 | 0.41 | 0.80 | 1.95 |
| Treatment 4 | 79.5 | 0.402 | 1.19 | 0.55 | 1.82 | 3.31 |
| Treatment 5 | 80.4 | 0.397 | 0.98 | 0.50 | 1.50 | 3.00 |
| Treatment 6 | 78.3 | 0.413 | 0.84 | 0.45 | 1.10 | 2.40 |

The $d_e$ of the treatment generated six measurements between 0.65 and 2.94 mm. The six UCs in the treatments were between 1.48 and 3.31. The porosity, $\varepsilon$, was in the range of 0.397–0.433.

Each treatment test (Figure 2) continued running for 14 days. During the test, water samples were collected and analyzed for TSS concentration. Details of the experiments carried out at different ranges of TSS, filtration velocity, head losses, and pollution load are given in Table 2.

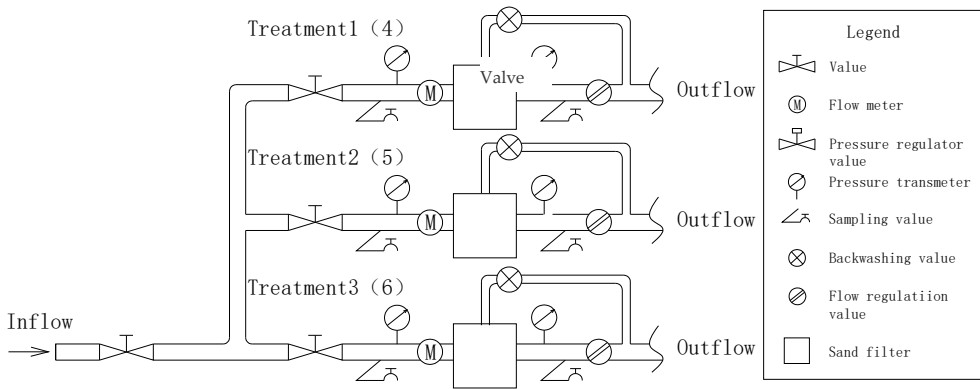

**Figure 2.** Sand filter pipe network showing the locations of the monitoring equipment.

**Table 2.** Range of variables during testing.

| Variable | Range | Mean and Standard Deviation |
|---|---|---|
| TSS $(\mathrm{kg}m^{-3})$ | 0.0035–0.088 | $0.0214 \pm 0.0152$ |
| Filtration velocity $(ms^{-1})$ | 0.0083–0.0398 | $0.03 \pm 0.007$ |
| Head loss (Pa) | 8750.00–166,517.85 | $55{,}584.47 \pm 32{,}777.10$ |
| Pollution load (Kg) | 0.0169–4.2049 | $0.91 \pm 0.919$ |

## 3. Model Building

*Generating Dimensionless Parameters*

The selection of model parameters affects the rationality and the predictive accuracy of the model. For example, in Table 3, the head loss calculation model typically involves three types of parameters, including filter structure parameters, sand media parameters, and parameters of the filtered liquid. Comparatively, the physical structure parameters were involved the least. Only the diameters of inlet and outlet pipes, $d_p$ [23], and the inner diameter of the sand filtration tank, $d_f$ [24], were involved. The structure parameters may be inferred in the calculation of the sand media parameters and the parameters of the filtered liquid. For example, there were no physical structure parameters involved in the theoretical model [25], but the inner diameter of the sand filtration tank $d_f$ was used to calculate the filtration velocity $v$. In the model proposed by Puig-Bargués et al. [22], the calculation of filtration surface area A required the use of parameters, such as the inner diameter of the sand filtration tank $d_f$ indirectly. The sand media parameters primarily included the equivalent diameter $d$, $d_e$, porosity $\varepsilon$, sphericity coefficient $\psi$, and total filtration surface area $A$. Characterization parameters (whether the equivalent diameter or effective diameter) of the sand media diameter were used more often, but porosity $\varepsilon$ and sphericity coefficient $\psi$ were only used in the theoretical model [25].

**Table 3.** Related technical parameters in previous studies.

| References | Tank Body Structure Parameters | Sand Media Parameters | Filtered Liquid Parameters |
|---|---|---|---|
| Dong, 1997 [25] | | Porosity $\varepsilon_0$<br>Equivalent diameter $d$<br>Sphericity coefficient $\psi$<br>Filtration layer thickness $L$ | Water density $\rho$<br>Water viscosity $\mu$<br>Acceleration of gravity $g$<br>Filtration velocity $v$ |
| Puig-Bargués et al., 2005 [22] | | Total filtration surface area $A$<br>Effective diameter $d_e$ | Solution density $\rho$<br>Water viscosity $u$<br>Volume of water filtered $V$<br>TSS concentration $C$<br>Flow rate of filtered liquid $Q$<br>Mean diameter of particle size distribution, $d_{\mathrm{p}}$ * |
| Duran-Ros et al., 2010 [23] | The filter inlet or the outlet diameter, $D_p$ | | Water density $\rho$<br>Water viscosity $u$<br>Filtration velocity $V$<br>the TSS concentration $C$ |
| Elbana et al., 2013 [17] | The internal diameter of the sand media filter tank $d_f$ | Effective diameter $d_e$ | Water density $\rho$<br>Acceleration of gravity $g$<br>Volume of water filtered $V$<br>TSS concentration $C$ |

\* The equation parameter $D_p$ in Puig-Bargués et al. [22] is changed to be $d_{\mathrm{p}}$ to differ from $D_p$ in Duran-Ros et al., (2010) [23].

$UC_s$ is an important characterization parameter of sand media [24]. $UC_s$ have not typically been used as model parameters because only one or two types of sand media

were used in the test in related studies. Therefore, it was difficult to identify the effects of different sand media $UC_s$ on head loss calculation. The parameters of the filtered liquid mainly involve liquid density $\rho$, water viscosity $\mu$, acceleration of gravity $g$, the TSS concentration $C$, the volume of filtered liquid $V$, and related parameters. The TSS concentration $C$ or the mean diameter of suspended solid $d_\mathrm{p}$ are usually used to characterize the special nature of the quality of reclaimed water; of these two parameters, the TSS concentration is used more often because it is easier to measure and usually has higher predictive accuracy [18,23,24]. $d_\mathrm{p}$ is rarely used because it is difficult to measure. In the model proposed by Puig-Bargués [18], there was an overlap between the volume of filtered liquid $V$ and the filtered liquid flow rate $Q$. The selected parameters in the dimensional analysis should be closely correlated while avoiding overlap. The removal of suspended particles by the sand media filter mainly depends on the effect of sedimentation and interception [19]. The head loss is affected primarily by the interception of suspended particles and by the media structure. Therefore, an accurate description of the physical characteristics of the sand-filtration media and the accumulation of suspended particles is critical for the accuracy of the head loss calculation model for sand media filters.

Based on previous studies, the $d_e$ (m), $d_{60}$ (m), filtration layer thickness $l$ (m), and sand mass m (kg) were first selected as the four characterization parameters of the sand filter in the paper. Six physical characterization parameters of liquid, including the water viscosity $\mu$ (Pa s), solution density $\rho$ (kg m$^{-3}$), acceleration of gravity $g$ (m s$^{-2}$), filtration velocity $v$ (m s$^{-1}$), pollution load $D$ (kg), and the rate of change of head loss $\Delta H$ (Pa), were selected.

$$D = \sum_{i=0}^{i=t} C_t Q_t \tag{4}$$

In this equation, $C_t$ is the TSS concentration (kg m$^{-3}$) in period $i$ in the backwash cycle period $t$. $Q_t$ is the volume of filtered liquid in period $i$ within backwash cycle $t$ (m$^{-3}$).

All ten parameters, the mass ($M$), and the dimensions of length ($L$) and time ($T$) were fit into a dimensionless matrix shown in Table 4. There were a total of ten ($m = 10$) independent variables and three basic variables ($k = 3$). Thus, there should be seven derived quantities as dimensionless parameters:

$$\pi_1 = \frac{v}{\sqrt{d_e g}} \cdot \pi_2 = \frac{d_{60}}{d_e} \cdot \pi_3 = \frac{l}{d_e} \, \pi_4 = \frac{m}{\rho d_e^3} \cdot \pi_5 = \frac{D}{\rho d_e^3} \cdot \pi_6 = \frac{\mu m}{D^{1.5} \rho^{0.5} g^{0.5}} \cdot \pi_7 = \frac{\Delta H}{d_e \rho g}$$

**Table 4.** Dimensional matrix for parameter selection.

|   | $\Delta H$ | $d_e$ | $d_{60}$ | $l$ | $m$ | $\rho$ | $g$ | $v$ | $\mu$ | $D$ |
|---|---|---|---|---|---|---|---|---|---|---|
| M | 1 | 0 | 0 | 0 | 1 | 1 | 0 | 1 | −1 | 1 |
| L | 1 | 1 | 1 | 1 | 0 | −3 | 1 | 0 | 1 | 0 |
| T | −2 | 0 | 0 | 0 | 0 | 0 | −2 | −1 | −1 | 0 |

The correlation among dimensionless parameters $\pi_1$ to $\pi_7$ was established based on Equation (5) to simulate and predict the head loss of a sand filter.

$$\pi_7 = f(\pi_1, \pi_2, \pi_3, \pi_4, \pi_5, \pi_6) \tag{5}$$

## 4. Statistical Analysis and Model Validation

The calculated or measured data of the flow rate and pressure and the measured TSS data were organized into a data series based on one-hour intervals according to the calculation equation of $\pi_1$ to $\pi_7$; these values were then converted into a corresponding logarithmic data series. Linear regression was performed using SPSS statistical software (SPSS Inc., Chicago, IL, USA) according to Equation (5). The significance level of the test

was set to 0.05, and the parameter exclusion criterion was $p > 0.05$. The results are shown in Table 5.

**Table 5.** Statistical analysis of the model.

| Dependent Variable | Significance Level $p$ | $R^2_{adj}$ | Independent Variable | Non-Standardized Coefficients | |
|---|---|---|---|---|---|
| | | | | B | Standard Deviation |
| $\ln \pi_7$ | <0.001 | 0.94 | Constants | 1.019 | 0.292 |
| | | | $\ln \pi_1$ | −0.192 | 0.083 |
| | | | $\ln \pi_2$ | −0.160 | 0.123 |
| | | | $\ln \pi_5$ | 0.636 | 0.021 |
| | | | $\ln \pi_6$ | 0.191 | 0.018 |

In Table 5, there are no non-standardized coefficients in $\ln \pi_3$ and $\ln \pi_4$, suggesting a significance level of correlation of $p > 0.05$. Therefore, $\ln \pi_3$ and $\ln \pi_4$ were excluded. The significance level of constants $\ln \pi_5$ and $\ln \pi_6$ in Table 6 was $p < 0.001$, and the significance level of $\ln \pi_1$ and $\ln \pi_2$ was $p < 0.05$, suggesting that the pollution load $D$ had more significant effects on dependent variables than the filtration velocity $v$ or particle diameter parameters $d_{60}$.

$$\frac{\Delta H}{d_e \rho g} = 2.77 \times \left( \frac{v}{\sqrt{d_e g}} \right)^{-0.192} \times \left( \frac{d_{60}}{d_e} \right)^{-0.160} \times \left( \frac{D}{\rho d_e{}^3} \right)^{0.636} \times \left( \frac{\mu m}{D^{1.5} \rho^{0.5} g^{0.5}} \right)^{0.191} \quad (6)$$

where $\frac{d_{60}}{d_e}$ is the $UC_s$. Thus, Equation (5) can be written as:

$$\frac{\Delta H}{d_e \rho g} = 2.77 \times \left( \frac{v}{\sqrt{d_e g}} \right)^{-0.192} \times UC_s^{-0.160} \times \left( \frac{D}{\rho d_e{}^3} \right)^{0.636} \times \left( \frac{\mu m}{D^{1.5} \rho^{0.5} g^{0.5}} \right)^{0.191} \quad (7)$$

**Table 6.** Statistical parameters comparison of calculated $\Delta H$ from equations and measured $\Delta H$, *Pa*.

| Statistical Parameters | Calculated Value by Equation (1) [a] | Calculated Value by Equation (2) [a] | Calculated Value by Equation (3) [a] | Calculated Value by Equation (7) [b] | Measured Value [b] |
|---|---|---|---|---|---|
| Maximum | $2.15 \times 10^{-24}$ | $1.83 \times 10^4$ | $6.19 \times 10^4$ | $1.52 \times 10^5$ | $1.67 \times 10^5$ |
| Minimum | $6.45 \times 10^{-292}$ | $1.24 \times 10^4$ | $9.57 \times 10^2$ | $9.06 \times 10^3$ | $8.75 \times 10^3$ |
| RMSE | $1.29 \times 10^{-26}$ | $8.04 \times 10$ | $1.13 \times 10^3$ | $2.12 \times 10^3$ | $2.54 \times 10^3$ |
| Mean | $1.29 \times 10^{-26}$ | $1.53 \times 10^4$ | $1.93 \times 10^4$ | $5.37 \times 10^4$ | $5.56 \times 10^4$ |
| Standard deviation | $1.66 \times 10^{-25}$ | $1.04 \times 10^3$ | $1.46 \times 10^4$ | $2.74 \times 10^4$ | $3.28 \times 10^4$ |
| Variation coefficient | 12.92 | 0.07 | 0.75 | 0.51 | 0.59 |

Note: [a] represents values with the same letters that are not significantly different ($p < 0.05$). [b] represents values with the same letters that are significantly different ($p < 0.05$).

In Figure 3, 3a is the correlation between the measured value of $\pi_7$ and the calculated value from Equation (7) ($p < 0.01$). Figure 3b shows that the residues have a relatively even distribution of approximately 0. Moreover, the significance level of the model reached $p < 0.001$ (as shown in Table 5), suggesting that the model derived with dimensional regression analysis could be used to calculate the head loss for a sand filter using reclaimed water in irrigation. The model may give satisfactory predictions within the range of operational and filter structure parameters, which is valid for the following conditions:

$$0.41 \leq d_e \leq 2.1 \text{ mm}; \ 1.48 \leq UC_s \leq 3.31; \ 0.0169 \leq D \leq 4.2049 \text{ kg}; 0.0038 \leq v \leq 0.0398 \ m \cdot s^{-1}$$

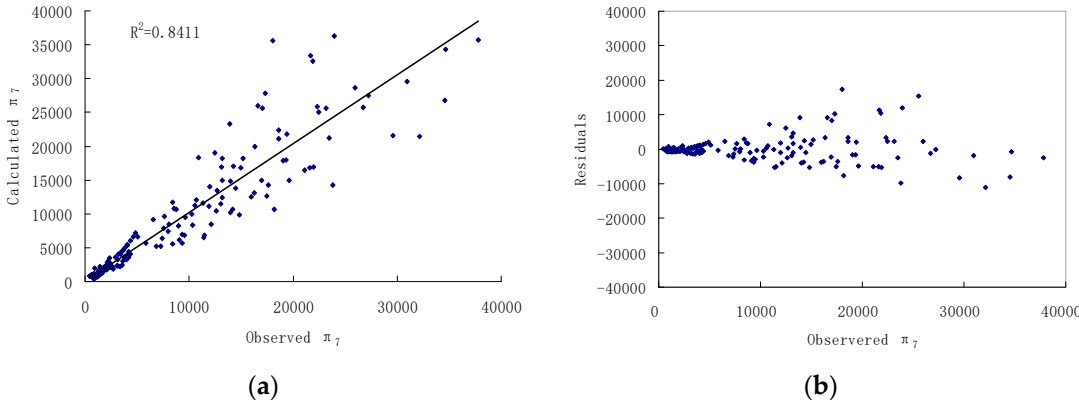

**Figure 3.** The measured value of $\pi_7$, the calculated value from Equation (6), (**a**) and residues (**b**).

## 5. Results and Discussion

Table 6 shows the statistical parameters of the calculated values from Equations (1)–(3) and (7) and the measured value of $\Delta H$. The measured values have no significant difference from the calculated values by Equation (6) and have a significant difference from the calculated values by Equations (1)–(3). The linear correlation coefficients between the calculated values from each model and the observed values were 0.009, 0.022, 0.146, and 0.643, respectively. The correlation coefficient of theoretical model 3 was higher than dimensional analysis models 1 and 2. The model built in this study, Equation (6), had the highest correlation. With respect to the mean values, the calculated values of Equations (2), (3) and (7) were in the identical order of magnitude as the measured value. Therefore, the statistical parameters in Table 1 suggest that the versatility of the dimensional analysis models and theoretical models found in the literature were poor because they all had certain limitations. The relatively large difference between statistical parameters of Equations (1)–(3) and measured values might be caused by factors such as the rationalities of parameter selection, quality differences in the water, and differences in sand media.

The process of dimensional analysis is to apply theorem $\pi$ to nondimensionalize the operational parameters of the phenomenon [28]. Therefore, it is important to identify the controlling parameters of the phenomenon. Compared to clean water, the most significant characteristic of reclaimed water is that it has a large number of suspended particles. In addition to the effects of the structural parameters of the filter and the physical characteristics of the sand-filtration media, the effects of the suspended particles in the reclaimed water should also be considered in the head loss calculation. It is critical to accurately characterize the number of suspended particles and the accumulation process. Equations (1) and (2) used concentration $C$ to characterize the concentration of suspended particles. Equations (2) and (7) used filtered water volume $V$ and pollution load $D$ to characterize the accumulated effect of the reclaimed water volume and the pollution load, respectively. Therefore, compared to Equation (1), it was more reasonable for Equation (2) to use the amount of filtered water as the value for the accumulated effect. Its correlation coefficient was also higher. Equation (3) represents the head loss calculation model for the clean filtration layer. Although no parameters characterizing the suspended particle amount and the accumulation process were used, this equation used parameters such as porosity $\varepsilon$, equivalent diameter $d$, sphericity coefficient $\psi$, and filtration layer thickness $L$ to accurately characterize the structure of the sand-filtration media. Because the removal of the suspended particles in reclaimed water mainly depends on the effect of sedimentation and interception [19], using $\varepsilon$ may therefore accurately characterize the effect of filtration media space on the interception of suspended particles and head loss. Therefore, the correlation coefficient of Equation (3) was much higher than that of Equations (1) and (2). Equation (7) used pollution load $D$ to characterize the effect of the suspended particle accumulation on head loss, which was more effective than the $TSS$ concentration or filtered water volume $V$

in the equation. In Table 1, $\ln \pi_5$ and $\ln \pi_6$ each included parameter $D$, and the fact that their significance level reached $p < 0.001$ supports this last point.

The difference in the quality of testing reclaimed water was also one of the factors affecting the accuracy of the model. The TSS concentration when testing reclaimed water was 4.4–18 g m$^{-3}$ for Equation (1) [23], 3.8–68.6 g m$^{-3}$ for Equation (2), and 3.5–88 g m$^{-3}$ for Equation (3), which was used to calculate the head loss of the clean filtration layer [24]. The TSS concentration differences suggest that there are differences in the number of suspended particles and their diameters; these differences further suggest that the differences in the removal efficiency of the filtration system [18] might cause differences in the sedimentation and interception of suspended particles in the pores of the sand media, resulting in head loss differences, which might cause the model to have low accuracy.

The effects of six different types of sand-filtration media on head loss were studied in this paper. The effective diameters of the media were 2.1 mm, 1.41 mm, 0.41 mm, 0.55 mm, 0.50 mm, and 0.45 mm. The ranges of $UC_s$ were 1.48, 1.56, 1.95, 3.31, 3.00, and 2.40, respectively. The testing sand media in Equation (1) had a $d_e$ of 0.27 mm and a $UC_s$ of 2.89 [23]. The testing sand media in Equation (2) had two diameters $d_e$ of 0.40 mm and 0.27 mm and a $UC_s$ of 2.41 and 2.89 [24]. Sand media with a diameter of 0.59 mm [25] were used in the test of Equation (3). $UC_s$ were introduced into Equation (7) as important parameters of the sand's physical characteristics and were also used with the other three parameters, $d_e$, sand mass (m), and filtration layer thickness (l), to characterize the key physical characteristics of the mass, volume, and particle uniformity of sand media, further improving the accuracy of the model and expanding the scope of its application.

## 6. Conclusions

The dimensional model Equation (7) was developed to predict head loss of reclaimed water for different types of sand-filtration media. In this model, three parameters, $UC_s$, $d_e$, and $m$, were used in Equation (7) to characterize the important physical characteristics of the mass, volume, and particle uniformity of sand media. In addition, the pollution load $D$ was used to characterize the effect of suspended particle accumulation on head loss. The correlation coefficient between the predicted and the measured head losses was 91.7%, and the significance level of the model reached $p < 0.001$. The results showed that the model might give satisfactory predictions within the range of operational and filter structure parameters, which is valid for the following conditions:

$$0.41 \leq d_e \leq 2.1 \text{ mm}; \ 1.48 \leq UC_s \leq 3.31; \ 0.0169 \leq D \leq 4.2049 \text{ kg}; 0.0038 \leq v \leq 0.0398 \ m \cdot s^{-1}$$

When compared to other models (Duran-Ros et al., 2010; Elbana et al., 2013), it was found that the poor versatility of the developed models for reclaimed water might be the result of factors such as the rationality of parameter selection, differences in water quality, and differences in sand-filtration media. The theoretical model [25] accurately described the structural characteristics of sand-filtration media without considering the characteristic of reclaimed water quality; the accuracy of the simulation remained higher than in Equations (1) and (2), suggesting the importance of selecting physical structure parameters of sand-filtration media. It is considered that the new model will also help enhance our knowledge of filtration and backwashing operations.

**Author Contributions:** Conceptualization, W.W.; methodology, W.W.; investigation, Y.H. (Yan Huang); Formal analysis, Y.H. (Yaqi Hu); Software, Y.H. (Yan Huang); Resources, H.L.; data curation, W.W.; writing—original draft preparation, W.W. and Y.H. (Yaqi Hu); writing—review and editing, X.B., R.L. and S.Y. All authors have read and agreed to the published version of the manuscript.

**Funding:** National Natural Science Foundation of China—Key Program 52079146.

**Institutional Review Board Statement:** Not applicable.

**Informed Consent Statement:** Not applicable.

**Data Availability Statement:** Data available upon request due to privacy and ethical restrictions.

**Acknowledgments:** The authors would like to express their gratitude to the editors and anonymous experts for their constructive comments.

**Conflicts of Interest:** The authors declare no competing financial interests.

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
