# Peer review of "Dimensional Analysis Model of Head Loss for Sand Media Filters in a Drip Irrigation System Using Reclaimed Water"

_water, doi:10.3390/w14060961_

Round 1

Reviewer 1 Report

A new equation for computing head loss in sand media filters was developed and compared with other equations which were previously obtained by other authors. Results are quite interesting, but manuscript needs to be improved.

Experiments should be more clearly described. Nothing is said about the number of replications of each treatment, but it is clear from Fig. 1 that the different treatments had different runs. It is important to fully describe the experiment since this allow other researchers to reproduce your experiment.

It seems that total head loss (maximum value of 167 kPa, according to Table 6) was noticeably higher than maximum head losses considered in the experiments used for obtaining previous equations. The effect of this different experimental procedure should be further discussed.

The authors should check head losses computed using equation (1) since these values seem too much different than others. Perhaps the authors are using this equation out of its range.

Tables and figures are wrongly enumerated. Moreover, some information is not properly sorted.

Specific comments:

  1. Page 1, line 8. The references of Liao et al. (2019a, 2019b) do not deal with using reclaimed effluent in irrigation. It seems an inappropriate self-citation made by the authors.
  2. It is not necessary to repeat the definition of those parameters (e.g., DH, C, v) with same symbol and units below equations (2) and (3).
  3. Page 3, lines 5-6. “As shown in Figure 1, three sand media filters were installed in the testing system”. Figure 1 shows measured values of P7, not the experimental setup. Figures must be presented correctly sorted.
  4. Figure 1. Besides R2, P-value of the regression of Fig. 1(a) should be shown.
  5. Page 3, line 7. Could an inverter measure pressure? Inverters are used for changing direct current to alternating current.
  6. Page 3. Table 5 should be renamed as Table 1 since it is the first table that appears in the manuscript.
  7. Page 3. Table. UCs values should have the same decimal positions.
  8. Page 3. “The test was divided into two sessions”. Please, provide more information about procedures and their possible effect on results.
  9. Page 3. It is not necessary to define again either de or UCs since they were previously defined after equations (2) and (3). So, those sentences related to de and UCs should be removed.
  10. The number of replications carried out for each treatment should be provided.
  11. Page 3. How was porosity measured? Nothing is said about it.
  12. Figure 2. For avoiding any confusion about the numbers within brackets, I would suggest changing “Treatment 1 (4)” with “Treatments 1 and 4” and similarly with the other treatments.
  13. Page 4. Section 3. Filtration surface area used in the quoted paper was superficial filtration area, and therefore depends on the inner diameter of sand filtration tank but not on the filtration layer thickness.
  14. Page 5. Table 1 should be renamed as Table 2 since it is the second table that appears in the manuscript.
  15. Page 6, below equation (4). The authors talk about the “backwash cycle period” but its meaning is unclear since no explanation about backwashing was provided in Materials and methods. Were these parameters only measured during backwashing? Were they only measured between backwashings (i.e., during each filtration run)? Please, reword and clarify it.
  16. Section 4. Significance level is the same as P. So, if exclusion criterion was P>0.05, the significance level should be 0.05.
  17. Page 6, below Table 3. A significance level P>0.05 cannot be suggested. Correlation has a P-value higher or lower than this threshold, but it is not suggested. Please, reword it.
  18. Page 7. Figure 1 (which should be renamed as Figure 2) should appear in this page.
  19. Page 7, Table 4. The legend “For values calculated by different equations, values having the same letters are not significantly different” is confusing since superscripts and b are showing significant differences between columns. It seems that significant differences were assessed for the mean, therefore letters for mean separation should appear with this parameter.
  20. Page 7, Table 4. Values computed using equation (1) are quite different than the others. The authors should check if there was any mistake and discuss these differences.
  21. Page 7, section 5. Are linear coefficients R2? Why are not they shown in Table 4?
  22. Page 7, section 5. According to table 4, should not be the model built in this study depicted with Equation 7 instead of equation 6?
  23. Page 7, section 5. There is not any statistical parameter shown in Table 5.
  24. Page 8, lines 1-4. The authors should also discuss the effect of the differences in the experimental setup on the results.
  25. Page 8, line 9. Change “tank” with “filter”.

Reviewer 2 Report

The paper: " Dimensional Analysis Model of Head Loss for Sand Media Filters in a Drip Irrigation System Using Reclaimed Water" of authors: Yaqi Hu Wenyong Wu, Yan Huang, Honglu Liu, Xiangshuai Bi, Renkuan Liao  and Shiyang Yin present a numerical investigation of the prediction of heat loss in sand media filters. Sand filters with six different media using reclaimed water were used to measure head losses at different flow rates in the laboratory. The parameters which were analysed are: uniformity coefficient, the effective diameter, the sand mass, the filtration velocity, the pollution load and the water viscosity.

This research could be interesting for the engineers involved in  chemical engineering as well as for the researcher in the field of wastewater treatment. this paper presents a attempt to develop a hydraulic model of a sand media filter using dimensional analysis through testing the hydraulic performance of the sand media filter

In my opinion, this paper can be published in your journal, and only MINOR revision is necessary.

Comments:

  1. In the conclusion section you can add a few sentences including the method and model used for this investigation.
  2. Did you develop your own computer code for numerical investigation?
  3. Maybe, you can add a few key words. Sometimes this can contribute to the citation index of the manuscript.

Author Response

This manuscript is a resubmission of an earlier submission. The following is a list of the peer review reports and author responses from that submission.